# Oct4 and Hypoxia Dual-Regulated Oncolytic Adenovirus Armed with shRNA-Targeting Dendritic Cell Immunoreceptor Exerts Potent Antitumor Activity against Bladder Cancer

**DOI:** 10.3390/biomedicines11102598

**Published:** 2023-09-22

**Authors:** Che-Yuan Hu, Chi-Feng Hung, Pi-Che Chen, Jia-Yu Hsu, Chung-Teng Wang, Ming-Derg Lai, Yuh-Shyan Tsai, Ai-Li Shiau, Gia-Shing Shieh, Chao-Liang Wu

**Affiliations:** 1Department of Urology, College of Medicine, National Cheng Kung University, Tainan 70101, Taiwanyouh@mail.ncku.edu.tw (Y.-S.T.); 2Department of Urology, Ditmanson Medical Foundation Chia-Yi Christian Hospital, Chiayi City 60002, Taiwan; 3Department of Biochemistry and Molecular Biology, College of Medicine, National Cheng Kung University, Tainan 70101, Taiwana1211207@mail.ncku.edu.tw (M.-D.L.); 4Department of Microbiology and Immunology, College of Medicine, National Cheng Kung University, Tainan 70101, Taiwanalshiau@mail.ncku.edu.tw (A.-L.S.); 5Ditmanson Medical Foundation Chia-Yi Christian Hospital, Chiayi City 60002, Taiwan; 6Department of Urology, Tainan Hospital, Ministry of Health and Welfare, Executive Yuan, Tainan 70043, Taiwan

**Keywords:** oncolytic adenovirus, dendritic cell immunoreceptor, immunotherapy, dendritic cell, Oct4, hypoxia, bladder cancer

## Abstract

Immunotherapy has emerged as a promising modality for cancer treatment. Dendritic cell immunoreceptor (DCIR), a C-type lectin receptor, is expressed mainly by dendritic cells (DCs) and mediates inhibitory intracellular signaling. Inhibition of DCIR activation may enhance antitumor activity. DCIR is encoded by *CLEC4A* in humans and by *Clec4a2* in mice. Gene gun-mediated delivery of short hairpin RNA (shRNA) targeting *Clec4a2* into mice bearing bladder tumors reduces DCIR expression in DCs, inhibiting tumor growth and inducing CD8^+^ T cell immune responses. Various oncolytic adenoviruses have been developed in clinical trials. Previously, we have developed Ad.LCY, an oncolytic adenovirus regulated by Oct4 and hypoxia, and demonstrated its antitumor efficacy. Here, we generated a *Clec4a2* shRNA-expressing oncolytic adenovirus derived from Ad.LCY, designated Ad.shDCIR, aimed at inducing more robust antitumor immune responses. Our results show that treatment with Ad.shDCIR reduced Clec4a expression in DCs in cell culture. Furthermore, Ad.shDCIR exerted cytolytic effects solely on MBT-2 bladder cancer cells but not on normal NIH 3T3 mouse fibroblasts, confirming the tumor selectivity of Ad.shDCIR. Compared to Ad.LCY, Ad.shDCIR induced higher cytotoxic T lymphocyte (CTL) activity in MBT-2 tumor-bearing immunocompetent mice. In addition, Ad.shDCIR and Ad.LCY exhibited similar antitumor effects on inhibiting tumor growth. Notably, Ad.shDCIR was superior to Ad.LCY in prolonging the survival of tumor-bearing mice. In conclusion, Ad.shDCIR may be further explored as a combination therapy of virotherapy and immunotherapy for bladder cancer and likely other types of cancer.

## 1. Introduction

Immunotherapy has recently become a promising modality for cancer treatment, mainly in active and passive forms. Passive immunotherapy is often used in patients with weaker immune systems or non-responsive or weakly responsive cancers. This type of treatment provides cell hormones, tumor-specific antibodies, or immune cells activated outside the body that can attack tumors in the body. The therapeutic effect is achieved by giving immunity extraneously. Active immunotherapy mainly aims to stimulate the patient’s internal immunity. It often uses the patient’s dendritic cells (DCs) as a carrier to deliver tumor antigens, oncolytic viruses, and immune checkpoint inhibitors, which can activate immune functions [1]. T-VEC (Talimogene laherparepvec) is a kind of oncolytic immunotherapy mediated by a modified herpes simplex virus type 1. It bolsters the antitumor immune response by selectively replicating within tumor cells and producing granulocyte-macrophage colony-stimulating factor (GM-CSF). T-VEC is the first immunotherapy proven to be effective against melanoma in phase III clinical trials and the first oncolytic virus approved by the FDA for melanoma patients [2]. Furthermore, combination treatment with oncolytic viruses and the immune checkpoint blockade anti-CTLA4 antibody has shown great promise in the treatment of various cancers [3].

DCs serve as antigen-presenting cells and are pivotal in the immune system. Once DCs uptake antigens and are activated, they present antigens to T cells to activate CD4^+^ and CD8^+^ T cells and thereby elicit immune responses [4]. The DC immunoreceptor (DCIR) expressed in various immune cells, such as DCs and macrophages, is encoded by *CLEC4A* (C-type Lectin Domain Family 4 Member) in humans and by *Clec4a2* in mice [5]. Clec4a2-deficient mice have more DCs and CD4^+^ T cells compared with wild-type mice. Furthermore, decreased DCIR expression in DCs promotes IL-12 production and enhances Th1 immunity. In addition, antigen-binding antibodies targeting DCIR increase CD8^+^ T cells through cross-presentation [6]. We have previously shown that downregulation of DCIR expression in DCs via gene gun-mediated delivery of *Clec4a* shRNA inhibits tumor growth and enhances CD8^+^ T cell immunity in the murine MBT-2 tumor model [7].

Adenovirus has been widely used as a vector for gene therapy. Two types of adenoviruses, replication-defective and replication-competent viruses, are applied. Replication-defective adenoviruses are used as viral vectors for gene therapy and vaccines. They are constructed by deletion of viral E1A or E1A/E1B genes and insertion of potent promoters to drive transgene expression in vitro and in vivo [3]. Numerous studies have shown that potent mammalian or viral promoters drive the expression of transgenes, typically delivering immune-related genes straight to tumor cells, thereby inducing local antitumor responses. Replication-competent adenoviruses produce viruses that replicate in and lyse cancer cells. Whether replication is defective or competent, adenoviruses produced can transmit or overexpress tumor-suppressive genes or cytotoxic genes in cancer cells, driving cell cycle arrest or cell apoptosis [8]. There are several advantages of using oncolytic adenoviruses for cancer therapy. Firstly, genetically modified viruses can replicate and kill within cancer cells without attacking normal cells. Secondly, oncolytic adenoviruses can quickly achieve high potency, making treatment more efficient. Lastly, adenoviruses do not integrate their genome into host chromosomes, thereby reducing the risk of insertional mutagenesis and increasing safety [9,10]. The first oncolytic adenovirus to undergo clinical trials was ONYX-015. The E1B-55 kD protein, which is essential for adenovirus replication in normal cells, facilitates the transportation of late gene mRNA from the nucleus to the cytoplasm. Furthermore, the E1B-55 kD protein can bind to p53, rendering it inactive. As a result, ONYX-015 was engineered to delete the E1B-55 kD gene. Infection of ONYX-015 in normal cells triggers a p53 response, halting cell growth or causing cell death. However, the virus can replicate massively in cells deficient in p53, which are frequently found in tumor cells, leading to lysing tumor cells [11].

Treatment with oncolytic adenoviruses primarily exploits their properties of specifically targeting and killing tumor cells. After infecting tumor cells, the viruses replicate massively and exert cytolytic effects on tumor cells. When the infected cells are lysed, many viruses, tumor cell antigens, and cell hormones are released, thereby attracting immune cells to infiltrate the tumor, such as DCs, natural killer cells, and T cells. These immune cells are activated and execute their functions within the tumor microenvironment. Additionally, oncolytic adenoviruses released after the tumor rupture continue to infect tumor cells and suppress tumor growth. In our previously published research, we constructed an oncolytic adenovirus driven by nine segments of the octamer-binding transcription factor 4 (Oct4) response element (ORE) in conjunction with six segments of the hypoxia response element (HRE), named Ad.LCY. Hypoxia-induced factors (HIF) 1 and 2α are highly expressed in the relatively hypoxic environment inside the tumor. It has also been suggested that HIF-2 regulates the expression of Oct4. Since HIF-1α and Oct4 are frequently overexpressed in the hypoxic environment of solid tumors, Ad.LCY can replicate efficiently in tumors. However, its replicative ability is weaker in an oxygen-rich environment compared to a hypoxic condition. We exploited these characteristics of Ad.LCY for bladder cancer treatment. Ad.LCY exhibits potent cytolytic effects against MBT-2 bladder tumor cells. Moreover, in the MBT-2 tumor model, Ad.LCY significantly inhibits tumor growth and improves the survival of the mice with established subcutaneous tumors [12].

Bladder cancer is one of the malignant diseases worldwide. In the early stages, surgery removes tumor tissues, followed by chemotherapy for bladder perfusion treatment to prevent cancer recurrence. If smaller superficial cancers or carcinoma in situ are not entirely removed after surgery, BCG can be used for bladder perfusion treatment. BCG enhances the immunity of patients by destroying bladder epithelial cells, thereby killing cancer cells [13]. The combination of surgery and BCG immunotherapy is the most effective treatment modality for non-muscle-invasive bladder cancer. Currently, immunotherapy is the direction of many treatment modalities. Previous studies have shown that bladder cancer patients with more immune cells infiltrating tumors and having a good prognosis also have a higher survival rate than patients with fewer infiltrating immune cells [14]. We have previously demonstrated that the oncolytic adenovirus Ad.LCY has a potent replicative ability in the hypoxic part of the tumor in treating mice with bladder tumors [12]. In the present study, we considered whether we could generate oncolytic adenoviruses carrying an immune regulation gene to achieve better treatment effects. DCs are antigen-presenting cells that present antigens to T cells for recognition and activation. Therefore, we focused on the regulation of DCs. Our previous study has indicated that downregulation of DCIR expression on DCs can achieve a tumor-suppressive effect and enhance CD8^+^ T cell immunity in the MBT-2 tumor model [7]. Therefore, the current study employed Ad.LCY as the oncolytic adenovirus carrying the shRNA specific to *Cleac4a* to investigate whether this novel oncolytic adenovirus could enhance antitumor immunity while retaining its tumor-selective cytolytic effect.

## 2. Materials and Methods

### 2.1. Cells and Mice

Murine MBT-2 bladder cancer cell line was obtained from Dr. Che-Rei Yang (Taichung Veterans General Hospital, Taichung, Taiwan), who originally obtained the cells from Roswell Park Memorial Institute (Buffalo, NY, USA). Murine NIH 3T3 cell line and human 293 embryonic kidney cell line were originally obtained from American Type Culture Collection (Manassas, VA, USA). Human A549 lung cancer cell line was purchased from the Bioresource Collection and Research Center (BCRC, Hsinchu, Taiwan). All cells were cultured in Dulbecco’s modified Eagle medium (DMEM) supplemented with 10% cosmic calf serum (Hyclone, Logan, UT, USA), 2 mM of L-glutamine, and 50 μg/mL of gentamicin. Female, 6- to 8-week-old C3H/HeN mice obtained from the Laboratory Animal Center of National Cheng Kung University were used for animal experiments.

### 2.2. Construction of Ad.shDCIR, a Derivative of Ad.LCY Expressing shRNA Targeting Clec4a

Different shRNA sequences specific to *Clec4a2* have been evaluated for downregulating the expression of Clec4a2 mRNA and protein [7]. We selected an effective shRNA sequence targeting *Clec4a* with high knockdown efficiency for constructing the oncolytic Ad.shDCIR adenovirus. The U6 promotor-shClec4a fragment of pLKO.1-shClec4a (TRCN0000077415) and the 6×HRE- 9×ORE-CMVmini fragment of pFRL2-6×HRE-9×ORE-CMVmini were amplified by a polymerase chain reaction (PCR) with two oligonucleotide primer pairs specific to U6-shDCIR (forward, 5′-GAGTAGAGTTTTCTCCTCGCATTCG ATTAGTGAACGGATC-3′; reverse, 5′-TTTGATGGAGGTCCCTGTAATAAA-CCCG-3′), and HRE-ORE-CMV (forward, 5′-TTACAGGGACCTCCATCAAAACAAAACG-3′; reverse, 5′-AGTCCCGGTGTCGGAGCGGCCTTCCATGGTGGCTTTAC-3′) [7,12]. These two DNA fragments were then ligated to the pShuttle vector, which had been digested with *Not*I and *Xol*I, via Gibson assembly to produce pShuttle-U6-shClec4a-6×HRE-9×ORE-CMVmini plasmid [15]. A new recombinant adenovirus was then generated using the Adeasy adenoviral vector system generously provided by Bert Vogelstein (John Hopkins School of Medicine) [16]. The pShuttle-U6-shClec4a-6×HRE-9×ORE-CMVmini was cut into a linear form with *Pme*I, and the DNA was then introduced into the *E. coli* BJ5183 strain carrying the Adeasy-1 plasmid. The strain with homologous recombination was selected with kanamycin, and the resultant recombinant plasmid was verified by digestion with *Pac*I. Finally, the completed oncolytic adenovirus plasmid was named pAd.shDCIR. The *Pac*I-digested pAd.shDCIR was transfected into the packaging 293 cells, and the formation of Ad.shDCIR was complete when the cells showed cytopathic effects (CPE). Oncolytic adenoviruses were propagated, purified, and quantified using the plaque assay in 293 cells.

### 2.3. Verification of Ad.shDCIR Expressing shRNA Targeting Clec4a

Although the plasmid size of pAd.shDCIR was confirmed with gel electrophoresis, it was unknown whether the recombinant Ad.shDCIR possessed inhibitory effects on silencing DCIR expression. We constructed a lentiviral vector carrying the coding sequence of Clec4a2-c-myc [7] and generated the corresponding recombinant lentivirus. A549 cells were transduced with recombinant lentiviruses encoding Clec4a2-c-myc fusion proteins to generate A549/Clec4a2-c-myc cells. The cells were infected with Ad.shDCIR for 48 h, and their DCIR expression was examined using an immunoblot analysis with anti-c-myc antibody.

### 2.4. Analysis of CPE and Cell Viability

For monitoring CPE, MBT-2 and NIH 3T3 cells were infected with either Ad.shDCIR or Ad.LCY at different multiplicities of infection (MOIs) or left uninfected (mock) under normoxic or hypoxic conditions for 48 h. The cells were then cultured in normoxic conditions for 5 days. To detect the CPE, cells were fixed and stained with 10% formalin/0.05% crystal violet. Cell viability was also determined with an MTS assay, using the CellTiter 96 AQueous One Solution cell proliferation assay kit (Promega, Madison, WI, USA).

### 2.5. Reverse Transcription Quantitative Real-Time PCR (RT-qPCR)

Trizol (Invitrogen, Carlsbad, CA, USA) was used to isolate total RNA from cells according to the manufacturer’s protocols. Total RNA (2 μg) was reverse-transcribed into cDNA using a Verso cDNA synthesis kit (Thermo Fisher Scientific, Barrington, IL, USA). The qPCR was performed using a SmartCycler System (Cephid, Sunnyvale, CA, USA) according to the manufacturer’s instructions. Each reaction contained 50 ng of cDNA, SYBR Premix Ex Taq (Takara Bio, Shiga, Japan) and 5 μmol of each forward and reverse primer. Three oligonucleotide primer pairs specific to DCIR (forward, 5′-CAAGAGTGAGGAGAACTGCTCC-3′; reverse, 5′-GCAGCATGAGTGTCCAAGATCC-3′), CD86 (forward, 5′-CATGGGCTTGGCAATCCTTA-3′; reverse, 5′-AAATGGGCACGGCAGATATG-3′), and human glyceraldehyde-3-phosphate dehydrogenase (GAPDH) (forward, 5′-GCCATCACTGCCACCCAG-3′; reverse, 5′-TCTTACTCCTTGGAGGCCATGT-3′) were used. Normalization was performed using GAPDH as an internal control, and relative gene expression was calculated using the comparative 2^(−ΔΔCt)^ method.

### 2.6. Assay of Cytotoxic T Lymphocyte (CTL) Activity

The in vitro CTL activity was assessed as described previously [7]. MBT-2/Luc cells that overexpress luciferase were used as target cells, whereas splenocytes harvested from Ad.shDCIR- or Ad.LCY-treated MBT-2 tumor-bearing mice served as effector cells. C3H/HeN mice were subcutaneously inoculated with MBT-2 cells (2 × 10^6^/site) in two sites into the flank of mice at day 0. Those mice with only one tumor growth were intratumorally administered with 2.5 × 10^8^ plaque-forming units (PFU) of Ad.shDCIR or Ad.LCY. Four days after adenoviral treatment, the mice were sacrificed, their spleens were collected, and single-cell suspensions were prepared. They were cultured in RPMI 1640 medium supplemented with 10% fetal bovine serum (Hyclone), 25 mM of HEPES, 2 mM of L-glutamine, and 50 μg/mL of gentamicin. The splenocytes were then treated with the MBT-2 cell lysate for 3 days for induction of CTL activities. Nonadherent cells were harvested and used as effector cells. Various numbers of the splenocytes (5 × 10^5^, 2.5 × 10^5^, 1.25 × 10^5^, and 6.25 × 10^4^) were plated in 96-well plates and cocultured with the target MBT-2/Luc cells (5 × 10^3^/well) in triplicate to achieve effector-to-target (E:T) ratios of 100:1, 50:1, 25:1, and 12.5:1, respectively. After incubation for 8 h at 37 °C, the cells were pelleted by centrifugation, and the supernatant was collected. Specific lysis was calculated based on the amount of luciferase released into the supernatant. The test supernatant (100 µL) was measured for luciferase activities with D-luciferin (Synchem, Felsberg, Germany) as the substrate using a luminometer (Lumat LB 9507, Berthold Technologies, Bad Wildbad, Germany). Light emission was recorded for 10 s.

### 2.7. Animal Studies

C3H/HeN mice were divided into three groups: the vehicle control (saline) group (*n* = 6), Ad.LCY group (*n* = 6), and Ad.shDCIR group (*n* = 7). MBT-2 cells (2 × 10^6^) were inoculated subcutaneously into the dorsal flank of the mice on day 0. All mice were monitored for tumor growth. Tumors were measured every three days in two perpendicular axes with a tissue caliper. Tumor volume was calculated as 1/6 π × (length of tumor) × (width of tumor)^2^ [17]. Treatments were begun when tumors reached an average volume of 100 mm^3^ (15 days). Tumor-bearing mice were treated intratumorally with three doses of 5 × 10^8^ PFU of Ad.LCY or Ad.shDCIR, or saline at days 15, 19, and 23. Tumor volumes were measured every three days. The time of animal death was recorded. All moribund mice, which were sacrificed, were also regarded as dead.

### 2.8. Statistical Analysis

All data are expressed as means ± standard error of the mean (SEM). Statistical significance between groups was assessed using one-way ANOVA. Differences in CTL activities and tumor volumes between groups were compared using repeated measures two-way ANOVA. The mouse survival was analyzed with the Kaplan–Meier survival curve and the log-rank test. Any *p*-value less than 0.05 is regarded as statistically significant.

## 3. Results

### 3.1. Ad.shDCIR Inhibits DCIR Expression in DCs

We first used A549/Clec4a2-c-myc cells to confirm the silencing effect of Ad.shDCIR. The expression of DCIR was reduced in cells infected with Ad.shDCIR as examined with an immunoblot analysis. Next, to ascertain whether Ad.shDCIR could efficiently inhibit DCIR expression in DCs, normal mouse bone marrow cells were collected and treated with GM-CSF (20 ng/ML) and IL-4 (10 ng/ML) for 5 days to induce differentiation into DCs. Bone-marrow-derived DCs (BMDCs) were exposed to the conditioned medium of MBT-2 cells that had been infected with oncolytic adenoviruses or were directly infected with oncolytic adenoviruses (Figure 1A). A notable reduction in DCIR expression was only seen in BMDCs treated with the conditioned medium obtained from MBT-2 cells infected with Ad.shDCIR (Figure 1B). This result suggests that oncolytic adenoviruses failed to replicate in DCs and thus could not deliver DCIR shRNA to silence DCIR expression. The conditioned medium from MBT-2 cells infected with Ad.shDCIR may contain functional shRNAs targeting DCIR (Figure 1B). CD86, which is constitutively expressed on DCs, Langerhans cells, macrophages, B cells, and other antigen-presenting cells, provides crucial costimulatory signals for T cell activation and survival. CD86 can induce signaling for self-regulation and cell–cell association or disassociation depending on the ligand bound. To further determine the activation status of DCs, we examined the expression of CD86 on BMDCs. Figure 1C shows that DCs were not significantly activated after sole infection with oncolytic adenoviruses. This reaffirms the difficulty of oncolytic adenoviruses in infecting and replicating in DCs, thereby failing to suppress DCIR expression. Nevertheless, after treatment with the conditioned medium from MBT-2 cells infected with oncolytic adenoviruses, which may contain many extracellular vesicles and could be engulfed by DCs, DCs were activated, and their CD86 expression was upregulated.

### 3.2. Ad.shDCIR Exhibits Cytolytic Effects on Mouse Bladder Cancer Cells but Not Normal Cells

Having demonstrated that Ad.shDCIR could inhibit DCIR expression, we further explored the specificity of its cytolytic activities against cancer cells. We compared the cytolytic effects of Ad.shDCIR and Ad.LCY on mouse MBT-2 bladder cancer cells and normal mouse NIH 3T3 fibroblasts. The two cell lines were infected with various doses of Ad.shDCIR or Ad.LCY for four days, and their survival was assessed with crystal violet staining and the colorimetric MTS assay. While Ad.LCY lysed MBT-2 cells at an MOI of 100, Ad.shDCIR at MOIs of 100 and even 50 lysed MBT-2 cells (Figure 2A,B), suggesting that Ad.shDCIR is superior to Ad.LCY in killing bladder cancer cells. Of note, these two viruses did not induce any cytolytic effects on NIH 3T3 cells, even at high MOIs (Figure 2C). Taken together, Ad.shDCIR can differentiate cancer cells from normal cells in its cytolytic activity.

### 3.3. The CTL Activity of Splenocytes from Tumor-Bearing Mice Treated with Ad.shDCIR Is Higher than That Treated with Ad.LCY

To explore whether knockdown of *Clec4a* expression delivered by oncolytic adenoviruses could induce antitumor immune responses, the CTL activity of mouse splenocytes obtained from MBT-2-tumor bearing mice that had been intratumorally administered with Ad.shDCIR or Ad.LCY for three times was evaluated (Figure 3A). Our results revealed that splenocytes from the mice receiving Ad.shDCIR induced a higher CTL response than those from mice receiving Ad.LCY (Figure 3B). We have previously performed the same CTL assay, which included effectors cells isolated from both the control mice and vector control mice [7]. Our results revealed that splenocytes from mice receiving shRNA targeting *Clec4a* induced stronger CTL responses than those from mice receiving a vector plasmid (pLKO_AS1) or no treatment (control). These two control splenocytes had similar basal levels of CTL activities. Since the present study is an extension of our previous work [7], we expect that the CTL activity of the two control splenocytes should be minimal.

### 3.4. Administration of Ad.shDCIR Inhibits Tumor Growth and Prolongs the Survival of Mice Bearing Established MBT-2 Tumors

Our cell studies discovered that the oncolytic adenovirus Ad.shDCIR could lyse tumor cells and suppress DCIR expression in DCs. To further explore whether downregulation of Clec4a2, a negative immune regulator of DCs, induced by the oncolytic Ad.shDCIR adenovirus could enhance its oncolytic efficacy, we compared the antitumor activities of Ad.shDCIR and Ad.LCY in terms of the tumor size and survival of mice bearing subcutaneous MBT-2 tumors. The protocol timeline is shown in Figure 4A. Our animal studies revealed delayed tumor growth in mice treated with either Ad.shDCIR or Ad.LCY compared to the vehicle (saline) control mice (Figure 4B). Interestingly, while tumor growth inhibition was comparable between the mice treated with Ad.shDCIR and Ad.LCY (Figure 4B), mice receiving Ad.shDCIR but not Ad.LCY significantly survived longer than those receiving saline. Therefore, Ad.shDCIR is superior to Ad.LCY in prolonging the survival of tumor-bearing mice.

## 4. Discussion

Prior research has shown that gene gun-mediated delivery of shRNA targeting *Clec4a2* to the skin slowed tumor growth in the MBT-2 bladder tumor model [7]. This treatment led to a noticeable increase in the infiltration of CD4^+^ and CD8^+^ T cells at the tumor site. Furthermore, splenocytes from mice that had received *Clec4a2* shRNA treatment exhibited a notably enhanced CTL activity compared to those from the control group. However, the antitumor effect induced by *Clec4a2* shRNA was abolished when CD8^+^ cells were depleted in mice [7]. In the present study, we further modified Ad.LCY, which is an Oct4 and hypoxia dual-regulated oncolytic adenovirus by inserting the sequence of *Clec4a2* shRNA driven by the U6 promoter upstream the 6×HRE-9×ORE-CMVmini promoter in the backbone of the adenoviral vector [12]. We thus created a novel oncolytic Ad.shDCIR adenovirus. DCs play a pivotal role in adaptive immunity, presenting antigens to naive T cells to trigger immune responses. Therefore, it is crucial for ensuring no cytolytic activity of Ad.shDCIR against DCs. Although DCs express lower levels of coxsackievirus and adenovirus receptor (CAR), integrins αvβ5 can promote the internalization of oncolytic adenoviruses but not viral attachment [18]. Therefore, oncolytic adenoviruses have low infectivity for DCs and usually require higher potency to achieve better infection efficiency. We did not observe a significant influence when we directly infected murine DCs with Ad.shDCIR. As oncolytic adenoviruses cannot replicate in normal cells, we presume that they have no noticeable effects on human DCs. Importantly, since Ad.shDCIR replicates preferentially in hypoxic tumor cells that overexpress Oct4, it exerts potent cytolytic activity against cancer cells while sparing normal cells, such as DCs [12].

DCs are crucial to adaptive immunity, capturing and presenting antigens to naive T cells. Conversely, immature DCs support immune suppression by inhibiting T cell activation. DCs also serve as immunomodulators and impact disease immunity. Clec4a, the mouse DCIR, is a lectin-2 family member expressed in various immune cells. Expression of Clec4a on DCs fluctuates according to their maturity. Since Clec4a is a negative immune regulator of DCs, mice deficient in Clec4a develop autoimmune diseases due to excess expansion of activated DCs and CD4^+^ T cells [19]. DCs expressing Clec4a inhibit inflammatory responses and T cell immunity during infections. In tuberculosis-infected models, DCIR-knockout DCs amplify IL-12 production, thereby fostering Th1 immunity. Moreover, DCIR-targeted antigen-conjugated antibodies can aid human CD8^+^ T cell priming through cross-presentation with DCs. Apoptotic cells can release numerous extracellular vesicles, possibly enclosing shClec4a from tumor cells, leading to DCIR suppression upon uptake with DCs. In the current study, BMDCs expressed reduced levels of DCIR after incubation with the conditioned medium from MBT-2 cells that had been infected with Ad.shDCIR. However, direct infection of BMDCs with Ad.shDCIR did not result in decreases in DCIR expression. There was also a trend in the downregulation of DCIR expression in BMDCs treated with the conditioned medium of uninfected MBT-2 cells. Since DCIR expression was reported to be downregulated upon DC activation, the DC activation marker CD86 was further examined. Notably, the conditioned medium of MBT-2 cells either infected with Ad.shDCIR or Ad.LCY, or without viral infection could upregulate CD86 expression indicative of DC activation. Moreover, BMDCs treated with the conditioned medium of Ad.shDCIR-infected MBT-2 cells appeared to express slightly lower CD86 levels compared to those treated with the conditioned medium of Ad.LCY-infected MBT-2 cells. Thus, our results show that Ad.shDCIR downregulates DCIR expression on DCs.

Hypoxia refers to a condition where there is a reduced level of oxygen supply in tissues or organs. This low-oxygen environment can affect cellular processes, including immune responses. Prolonged or excessive hypoxia can impair DC maturation [20]. It has been observed that chronic hypoxia can lead to the accumulation of immature or partially mature DCs, which are less capable of initiating effective immune responses. This impaired maturation can result in a reduced ability to prime T cells and generate robust immune reactions [21]. Furthermore, hypoxia can modulate other factors involved in DC function, like the production of immunosuppressive molecules, such as vascular endothelial growth factor and IL-10. These molecules can further dampen the maturation of DCs and immune-stimulating capacity [21].

Glucose and glycolysis are crucial for inflammatory functions of many immune cells, and depletion of glucose in pathological microenvironments is related to defective immune responses. Glucose has an opposite effect on DCs, suppressing the inflammatory output of lipopolysaccharide-stimulated DCs and inhibiting T-cell responses induced by DCs [22]. In hypoxic DCs, expression of HIF-1α significantly increases. Furthermore, in hypoxic conditions, expression of CD86 and CD40 on DCs decreases, while expression of PD-L1 and LAG-3, an inhibitory immune checkpoint molecule comparable to PD-1 and CTLA-4, increases. However, the HIF-1α inhibitor HIPX478 restores the maturity of DCs and weakens the expression of immune suppressive molecules [23]. Moreover, overexpression of HIF-1α prevents the maturity of DCs and reduces the tumor-killing ability of effector T cells, induces the differentiation of regulatory T cells, and accelerates T cell exhaustion. Therefore, inhibition of HIF-1α effectively prevents immune suppression and improves antitumor immunity [24].

Oct4 is a transcription factor that plays a critical role in maintaining pluripotency and self-renewal of embryonic stem cells. Its expression is upregulated in certain types of cancer cells, contributing to their stemness, proliferation, and resistance to therapies. The primary role of Oct4 is therefore related to stem cells and cancer cells. In cancer biology, the interaction between Oct4 and the immune system may occur indirectly. Cancer cells with high Oct4 expression can potentially manipulate the immune system to create a more immunosuppressive tumor microenvironment, thereby promoting tumor immune evasion [25].

Tumor hypoxia correlates with increased PD-L1 expression in the tumor tissue of bladder cancer [26]. PD-L1, an immune checkpoint inhibitor, has a well-documented anticancer therapeutic effect in most types of cancers. PD-L1 expression is strongly associated with stem cell markers in cancer cells. Inhibition of PD-L1 with immune checkpoint inhibitors might decrease the stem cell population that is known to be associated with cancer recurrence [27]. Expression profiling of cancer datasets has shown a statistically significant correlation between PD-L1 expression and stemness score. PD-L1 through AKT activation positively affects Oct4 expression, and downregulation of PD-L1 compromises the self-renewal capability of cancer stem cells [28]. We have previously demonstrated that either Ad.9OC (an Oct4-dependent oncolytic adenovirus driven by nine copies of the ORE) or Ad.LCY (an Oct4 and hypoxia dual-regulated oncolytic adenovirus driven by the HRE/ORE hybrid promoter) exerts oncolytic activities in mice bearing human TCCSUP tumor xenografts or murine MBT-2 syngeneic tumors. Notably, Ad.LCY exhibits higher antitumor activity than Ad.9OC in NOD/SCID mice bearing human TCCSUP bladder tumor xenografts in terms of inhibiting tumor growth and prolonging mouse survival [12]. In C3H/HeN mice bearing murine MBT-2 syngeneic tumors, Ad.LCY is more effective in inhibiting tumor growth; however, the two viruses exert similar effects on extending mouse survival [12]. Since these two oncolytic adenoviruses exploit tumor properties of Oct4 overexpression and hypoxia for designing their tumor selectivity and efficacy, they exert similar antitumor effects in immunocompetent mice [12]. Xenograft mouse models of human tumors grown in immunocompromised mice, such as NOD/SCID mice, are widely used for studying cancer metastasis and anticancer drug screening. The xenograft model does not mimic the natural immune response of the human body due to immunodeficiency. By contrast, syngeneic tumor models in immunocompetent mice provide the appropriate tumor microenvironment and immunologic compatibility between tumor cells and the mice, which is more closely representative of the mouse tumor and more closely exemplifies human physiology. In the current study, we employed strategies of tumor-selective virotherapy and silencing of the negative immune regulator Clec4a to generate Ad.shDCIR. Since this novel engineered adenovirus combines oncolytic virotherapy and immunotherapy, it is more appropriate to use a syngeneic MBT-2 tumor model in immunocompetent C3H/HeN mice for evaluating its antitumor efficacy.

In the present study, while Ad.shDCIR and Ad.LCY exert similar antitumor effects on retarding tumor growth, Ad.shDCIR is superior to Ad.LCY in prolonging mouse survival. These seemingly contradictory results deserve further investigation. However, it is not necessarily expected that tumor-bearing mice would definitely die of larger tumor burdens. Furthermore, the mice may also die as a result of metastatic tumors even in small sizes. Thus, it is reasonable to speculate that Ad.shDCIR may be more potent than Ad.LCY in promoting antitumor immune responses, which may be beneficial for inhibiting tumor metastasis, leading to enhancing the survival of tumor-bearing mice. However, further studies are required to elucidate the mechanisms of action of these oncolytic adenoviruses.

In conclusion, we have developed Ad.shDCIR, an Oct4 and hypoxia dual-regulated oncolytic adenovirus armed with *Clec4a* shRNA. Hypoxia, stemness, and immunosuppression are hallmarks of tumor growth and progression. Given that Ad.shDCIR can be used as a combination therapy of virotherapy and immunotherapy, it may be further exploited as a broad-spectrum anticancer agent.

## Figures and Tables

**Figure 1 biomedicines-11-02598-f001:**
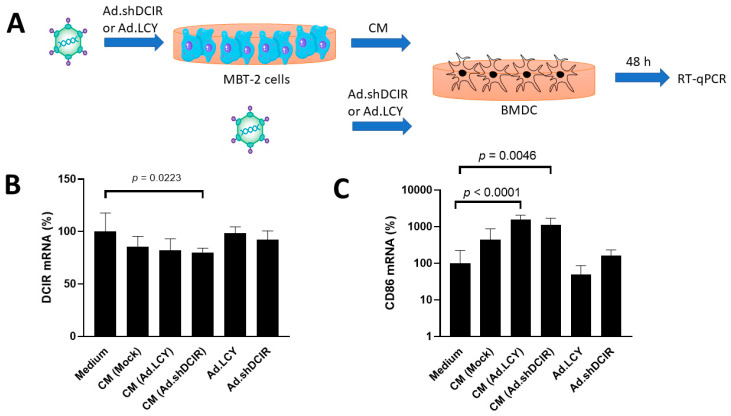
Expression levels of DCIR and CD86 on DCs after treatment with the conditioned medium of MBT-2 cells infected with oncolytic adenoviruses or directly infected with oncolytic adenoviruses. (**A**) BMDCs (1 × 10^5^) cultured in 6-well plates with three replicates for each group were either directly infected with Ad.DCIR or Ad.LCY at an MOI of 100 or treated with the conditioned medium (CM) of MBT-2 that had been infected with Ad.DCIR or Ad.LCY at an MOI of 100. After 48 h, total RNA from BMDCs was extracted and analyzed for DCIR and CD86 expression using RT-qPCR. (**B**,**C**) The mRNA expression levels of DCIR (**B**) and CD86 (**C**) on DCs. The expression level of control cells in culture medium was arbitrarily set to 100. Statistical analysis is performed using one-way ANOVA.

**Figure 2 biomedicines-11-02598-f002:**
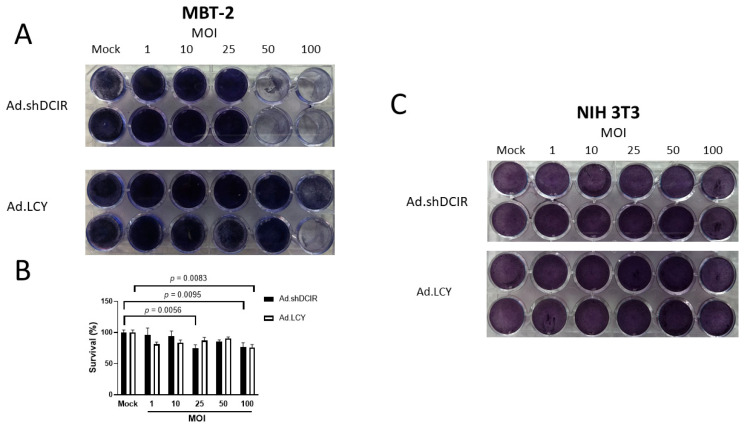
Differential cytolytic effects of Ad.shDCIR on MBT-2 bladder cancer cells and normal NIH 3T3 fibroblasts. (**A**) MBT-2 cells (1 × 10^5^) were cultured in 24-well plates (two replicates per group) and infected with Ad.shDCIR or Ad.LCY at different MOIs. After four days, they were stained with crystal violet for observing CPE. (**B**) MBT-2 cells (1 × 10^3^) were cultured in 96-well plates and infected with Ad.shDCIR at different MOIs (four replicates per group). After three days, cell viability was examined with the MTS assay. Statistical analysis was performed using one-way ANOVA. (**C**) NIH 3T3 cells (1 × 10^5^) were cultured in 24-well plates (two replicates per group) and infected with Ad.shDCIR or Ad.LCY at different MOIs. After four days, they were stained with crystal violet. Mock-treated cells corresponded to uninfected cells in culture medium.

**Figure 3 biomedicines-11-02598-f003:**
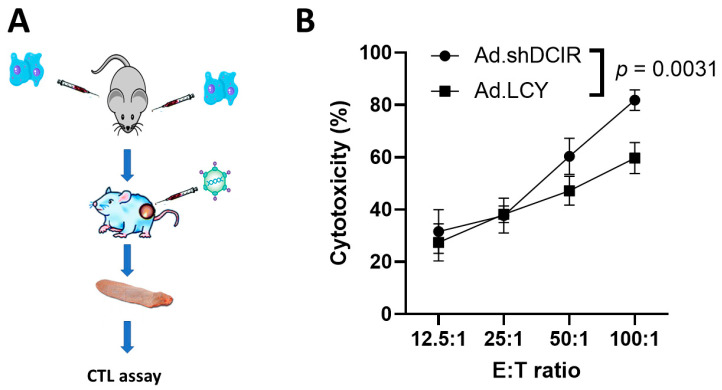
The CTL activity is significantly higher in tumor-bearing mice treated with Ad.shDCIR than in those treated with Ad.LCY. (**A**) Experimental design. Mice were subcutaneously inoculated with MBT-2 (2 × 10^6^/site) in two sites into the flank of mice on day 0. Those mice with only one tumor growth received intratumoral injection of 2.5 × 10^8^ PFU of Ad.shDCIR or Ad.LCY. Four days after adenoviral treatment, splenocytes serving as effector cells were assessed for their CTL activities against MBT-2/Luc cells serving as target cells. (**B**) The CTL activities of tumor-bearing mice receiving Ad.shDCIR or Ad.LCY against MBT-2/Luc cells were measured via detection of the release of luciferase into the supernatant after cell death (mean ± SEM; *n* = 3).

**Figure 4 biomedicines-11-02598-f004:**
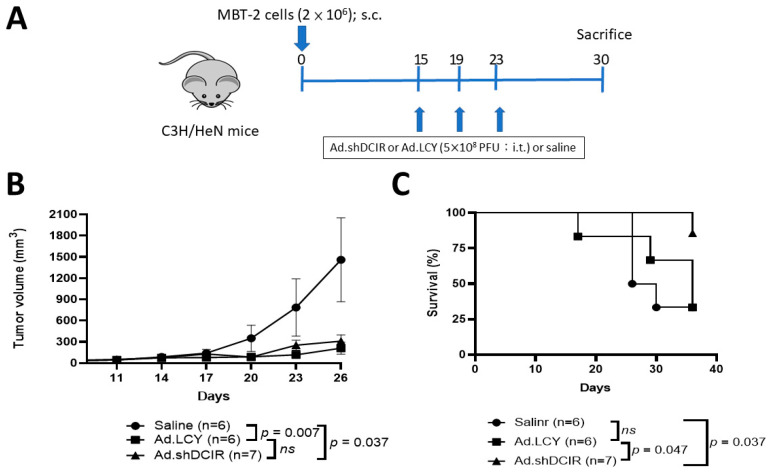
Antitumor activities of Ad.shDCIR and Ad.LCY against syngeneic MBT-2 tumors. (**A**) Treatment schedule. Groups of female C3H/HeN mice (*n* = 6~7) were subcutaneously inoculated with 2 × 10^6^ MBT-2 cells on day 0. The mice were intratumorally injected with three doses of 5 × 10^8^ PFU of Ad.LCY or Ad.shDCIR, or the vehicle (saline) on days 15, 19, and 23. (**B**) Tumor volumes were measured every three days, and mean tumor volumes in each group are shown. (**C**) Kaplan–Meier survival curves for each group are shown. Statistical analyses were conducted using repeated measures two-way ANOVA to compare tumor growth (**B**) and the log-rank test to compare survival time (**C**).

## Data Availability

The original data are available from the corresponding authors upon reasonable request.

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
