# Peer review of "Oct4 and Hypoxia Dual-Regulated Oncolytic Adenovirus Armed with shRNA-Targeting Dendritic Cell Immunoreceptor Exerts Potent Antitumor Activity against Bladder Cancer"

_biomedicines, 2023, doi:10.3390/biomedicines11102598_

Round 1

Reviewer 1 Report

Comments 1. Some phrasing is not clear or confusing, especially in discussion.

2. Methods: a. the amount of cells used for lysates in CTL assays must be determined

(related to  fig3). Control spleen cells must be included.

b. Number of mice  per group must be provided (related to  fig 4). 

c. Tumor volume formula must be checked

d. Mock infections related to all figures-results   must include PBS and an empty adenovirus control. Are the viruses replicative eor not in the different cell types?

3. Results :  Fig 1:  They measure RNA, but protein levels are more informative for this king of experiment.  The conclusion that   DCs ate not directly infectable by adenovirus is important.  Is this true for human DC also.   The presence of extracellular vesicles carrying the shRNA of interest is interesting but they do not provide evidence.

Fig2. In A,B, C , which cells are NIH3T3 and which MBT2 is confusing. The replicates are not clearly shown.

Fig4. Where the legend, statistics, number of mice of part C  (survival) of this figure? Results of B and C concerning Ad.LCY are contradictory.   Since the LCY tumors hardly grow  (similar to  DCIR that however has good survival) what is the cause of death?

Discussion an general experimental set up :1.  The part of  human xenogrfats vs   isogeneic mouse grafts is confusing.

2. To determine if there is an immune component in what they observed they should perform parallel experiments in immunocompromised mice.

3. Their data  may simply reflect an empty virus effect in cell survival

Needs improvement 

Author Response

Oct4 and hypoxia dual-regulated oncolytic adenovirus armed with shRNA targeting dendritic cell immunoreceptor exerts potent antitumor activity against bladder cancer

Hu et al. (biomedicines-2540399)

Point-by-point responses to the editor and reviewers:

Response to the editor:

We have reworded the parts indicated by repetition check.

Comments 1. Some phrasing is not clear or confusing, especially in discussion.

Response:

We apologize for lack of clarity and causing confusion We have carefully revised the manuscript on the basis of science and English language.

  1. Methods:
  2. the amount of cells used for lysates in CTL assays must be determined

Response:

Thank you for your helpful suggestion. We have added the required information for the CTL assay. We included the following in Materials and Methods section in the revised version of the manuscript (page 5, lines 210-214).

“Nonadherent cells were harvested and used as effector cells. Various numbers of the splenocytes (5 × 105, 2.5 × 105, 1.25 × 105, and 6.25 × 104) were plated in 96-well micro-plates and cocultured with the target MBT-2/Luc cells (5 × 103/well) in triplicate to achieve effector-to-target (E:T) ratios of 100:1, 50:1, 25:1, and 12.5:1.”

(related to  fig3). Control spleen cells must be included.

Response:

Thank you for your helpful comment. Since the purpose of the CTL assay shown in Figure 3B was to compare the antitumor activity of the spleen cells collected from tumor-bearing mice that had been administered with Ad.shDCIR or Ad.LCY. The assay did not include control spleen cells obtained from normal mice. We presume that the CTL activity of control splenocytes should be minimal.    

  1. Number of mice  per group must be provided (related to  fig 4). 

Response:

We have added the mouse number per group in the Materials and Methods section (page 5, lines 221-222).

  1. Tumor volume formula must be checked

Response:

To enhance clarity, we have revised the formula for measuring tumor volumes in the Materials and Methods section (page 5, lines 224-226). We stated “Tumors were measured every three days in two perpendicular axes with a tissue caliper. Tumor volume was calculated as (length of tumor) × (width of tumor)2 × 0.45.”

  1. (1) Mock infections related to all figures-results  must include PBS and an empty adenovirus control.

Response:

Thank you for your critical comment. In Figures 1B and 1C, culture medium was used as the control. In Figure 2, mock-treated cells corresponded to uninfected cells in culture medium. In Figure 4, saline served as the vehicle control. However, in Figure 3, we did not include naïve normal splenocytes as control effector cells. The backbones of oncolytic adenoviruses, such as Ad.shDCIR and Ad.LCY, were very different from those of wild-type adenoviruses. Oncolytic adenoviruses can selectively replicate in and kill tumor cells while sparing normal cells. Wild-type adenovirus can infect a variety of normal and tumor cells, leading to cell death. Therefore, this unmodified wild-type adenovirus is not suitable to serve as control viruses. Of note, the backbone of Ad.shDCIR and Ad.LCY was identical except for the expression cassette of DCIR shRNA in Ad.shDCIR. Therefore, the two oncolytic adenoviruses are appropriate for side-by-side comparisons for the advantage of shDCIR expression in enhancing their antitumor activity.

(2) Are the viruses replicative or not in the different cell types?

Response:

We have previously published several papers regarding the construction of different E1B-55 kD-deleted oncolytic adenoviruses with various promoters or response elements that are active in cancer cells but not in normal cells. These engineered adenoviruses replicate in and lyse different human bladder cancer cells,1 but not normal human urothelial cells.2,3 Importantly, we have also identified that murine MBT-2 cancer cells are permissive to adenoviral infection.4 Notably, these oncolytic adenoviruses can replicate in MBT-2 cells. Furthermore, adenoviral fiber and hexon proteins, which are two viral late protein, are detected in tumor tissue of the mice receiving oncolytic virotherapy, indicating oncolytic adenoviruses can produce viral progeny in MBT-2 cells and tumors.2,3 Therefore, MBT-2 tumor model allows us to evaluate the therapeutic efficacy of oncolytic adenoviruses in immunocompetent mice. Compared to xenograft tumor models in immunodeficient mice, syngeneic tumor models in immunocompetent mice provide the appropriate tumor microenvironment and immunologic compatibility between tumor cells and the animal that is more closely representative of the mouse tumor and also more closely exemplifies the human physiology.

References:

  1. Hsieh JL, Wu CL, Lai MD, Lee CH, Tsai CS, Shiau AL. Gene therapy for bladder cancer using E1B-55 kD-deleted adenovirus in combination with adenoviral vector encoding plasminogen kringles 1-5. Br J Cancer. 2003; 88:1492-9. PMID: 12778082.
  2. Chang CC, Shieh GS, Wu P, Lin CC, Shiau AL, Wu CL. Oct-3/4 expression reflects tumor progression and regulates motility of bladder cancer cells. Cancer Res. 2008; 68:6281-91. PMID: 18676852.
  3. Wu CL, Shieh GS, Chang CC, Yo YT, Su CH, Chang MY, Huang YH, Wu P, Shiau AL. Tumor-selective replication of an oncolytic adenovirus carrying oct-3/4 response elements in murine metastatic bladder cancer models. Clin Cancer Res. 2008; 14:1228-38. PMID: 18281558.
  4. Shieh GS, Shiau AL, Yo YT, Lin PR, Chang CC, Tzai TS, Wu CL. Low-dose etoposide enhances telomerase-dependent adenovirus-mediated cytosine deaminase gene therapy through augmentation of adenoviral infection and transgene expression in a syngeneic bladder tumor model. Cancer Res. 2006; 66:9957-66. PMID: 17047058.

  1. Results :

Fig 1:  They measure RNA, but protein levels are more informative for this king of experiment.  

Response:

In Figure 1, we detected mRNA levels of DCIR in bone marrow-derived dendritic cells (BMDCs) after treatment with the conditioned medium of MBT-2 cells that had been infected with oncolytic adenoviruses. Since shRNAs generally interact with the 3’-UTR regions of their targets, resulting in mRNA cleavage, we examined the effect of Ad.shDCIR on downregulation of Clec4a mRNA expression with RT-qPCR. Thus, we did not detect the protein levels of the treated cells. Nevertheless, we have used A549/DCIR-c-myc cells to examine Clec4a protein levels in cells after infection with Ad.shDCIR by immunoblot analysis with anti-c-myc antibody (Table 1). We have added the following in the Results section of the revised version of the manuscript (page 5, lines 241-243).

“We first used A549/DCIR-c-myc cells to examine the silencing effect of Ad.shDCIR. The expression of DCIR was reduced in cells infected with Ad.shDCIR as examined by immunoblot analysis (data not shown).”

Table 1. Knockdown efficacy of Ad.shDCIR in reducing DCIR-c-myc protein expression. A549 cells were transduced with lentiviral vectors encoding DCIR-c-myc fusion proteins to generate A549/DCIR-c-myc cells. The cells were infected with Ad.shDCIR for 48 h, and their DCIR expression was examined by immunoblot analysis with anti-c-myc antibody.

The conclusion that DCs ate not directly infectable by adenovirus is important.  Is this true for human DC also.  

Response:

Since Ad.shDCIR cannot replicate and induce cytolytic effects in normal cells, it fails to silence DCIR expression in DCs. As oncolytic adenoviruses cannot replicate in normal cells, we presume that they have no noticeable effects on human DCs.

The presence of extracellular vesicles carrying the shRNA of interest is interesting but they do not provide evidence.

Response:

Thank you for your constructive comment. We have added the following in the Discussion section of the revised version of the manuscript (page 9, lines 374-379):

“In the present study, we did not provide evidence to show that extracellular vesicles may carry shRNAs of interest. However, there are accumulating data showing that extracellular vesicles can be released almost ubiquitously by every living cell, including tumor cells and immune cells. They can carry various molecules, such as protein, DNA, RNA, microRNA, long non-coding RNA (lncRNA), shRNA, and more. The extracellular vesicles play roles in cancer progression and immune regulation.”

Fig2. In A,B, C , which cells are NIH3T3 and which MBT2 is confusing. The replicates are not clearly shown.

Response:

We have revised the legend of Figure 2 and indicated MBT-2 and NIH 3T3 cells in A and C panels of Figure 2, respectively, in the revised version of the manuscript.

Fig 4. Where the legend, statistics, number of mice of part C  (survival) of this figure?

Response:

The legend for Figure 4 has been revised in the revised version of the manuscript (page 8, lines 331-338).

Results of B and C concerning Ad.LCY are contradictory. Since the LCY tumors hardly grow  (similar to  DCIR that however has good survival) what is the cause of death?

Response:

Thank you for raising this important question. We have added the following in the Discussion section of the revised version of the manuscript (page 11, lines 454-462):

“In the present study, while Ad.shDCIR and Ad.LCY exerted similar antitumor effects on retarding tumor growth, Ad.shDCIR was superior to Ad.LCY in prolonging mouse survival. These seemingly contradictory results deserve further investigation. However, it is not necessarily expected that tumor-bearing mice would definitely die of larger tumor burdens. Furthermore, the mice may also die as a result of metastatic tumors even in small sizes. Thus, it is reasonable to speculate that Ad.shDCIR may be more potent than Ad.LCY in promoting antitumor immune responses, which may be beneficial for inhibiting tumor metastasis, leading to enhancing the survival of tumor-bearing mice. However, further studies are required to elucidate the mechanisms of action of these oncolytic adenoviruses.”    

Discussion an general experimental set up :

  1. The part of  human xenografts vs isogeneic mouse grafts is confusing.

Response:

Thank you for your critical comment. We have revised the text in the Discussion section in the revised version of the manuscript as follows (page 10, lines 431-442):

“We have previously demonstrated that either Ad.9OC (an Oct4-dependent oncolytic adenovirus driven by nine copies of the ORE) or Ad.LCY (an Oct4 and hypoxia dual-regulated oncolytic adenovirus driven by the HRE/ORE promoter) exerts oncolytic activities in mice bearing human TCCSUP tumor xenografts or murine MBT-2 syngeneic tumors. Notable, Ad.LCY exhibits higher antitumor activity than Ad.9OC in NOD/SCID mice bearing human TCCSUP bladder tumor xenografts in terms of inhibiting tumor growth and prolonging mouse survival [12]. In C3H/HeN mice bearing murine MBT-2 syngeneic tumors, Ad.LCY is more effective in inhibiting tumor growth; however, the two viruses exert similar effects on extending mouse survival [12]. Since these two oncolytic adenoviruses exploit tumor properties of Oct4 overexpression and hypoxia for designing their tumor selectivity and efficacy, they exert similar antitumor effects in immunocompetent mice [12].”

  1. To determine if there is an immune component in what they observed they should perform parallel experiments in immunocompromised mice.

Response:

Thank you for your valuable comment. We have included the text in the Discussion section of the revised version of the manuscript as follows (page 10, lines 442-453).

“Xenograft mouse models of human tumors grown in immunocompromised mice, such as NOD/SCID mice, are wildly used for studying cancer metastasis and anticancer drug screening. The xenograft model does not mimic the natural immune response of the human body due to immunodeficiency. By contrast, syngeneic tumor models in immunocompetent mice provide the appropriate tumor microenvironment and immunologic compatibility between tumor cells and the mice that is more closely representative of the mouse tumor and also more closely exemplifies human physiology. In the current study, we employed strategies of tumor-selective virotherapy and silencing of the negative immune regulator Clec4a to construct Ad.shDCIR. Since this novel engineered adenovirus combines oncolytic virotherapy and immunotherapy, it is more appropriate to use syngeneic MBT-2 tumor model in immunocompetent C3H/HeN mice for evaluating its antitumor efficacy.

  1. Their data  may simply reflect an empty virus effect in cell survival

Response:

Thank you for your valuable comment. Originally, we constructed Ad5WS1, which is an E1B-55 kD-deleted oncolytic adenovirus derived from human adenovirus type 5, driven by the adenoviral E1A promoter.1 Subsequently, we replaced the E1A promoter in the viral backbone with the Oct4 response element (ORE) and the hypoxia response element (HRE) in conjunction with the ORE  to generate Ad.9OC and Ad.LCY targeting Oct4 (stemness) and hypoxia in tumors, respectively.2,3 In the present study, we further inserted the expression cassette of shClec4a driven the U6 promoter upstream the HRE/ORE hybrid promoter to create Ad.shDCIR. Considering the complexity of these engineered adenoviruses, it is not suitable to use an empty virus, either wild-type human adenovirus type 5 or replication-defective adenoviral vector frequently used as a gene therapy or vaccine vector, as the control virus. On the basis of our previous reports, our oncolytic adenoviruses indeed exert tumor-selective cytolytic activities in vitro and oncolytic activities in mouse tumor models.    

References:

  1. Hsieh JL, Wu CL, Lee CH, Shiau AL. Hepatitis B virus X protein sensitizes hepatocellular carcinoma cells to cytolysis induced by E1B-deleted adenovirus through the disruption of p53 function. Clin Cancer Res. 2003; 9:338-45. PMID: 12538486.
  2. Wu CL, Shieh GS, Chang CC, Yo YT, Su CH, Chang MY, Huang YH, Wu P, Shiau AL. Tumor-selective replication of an oncolytic adenovirus carrying oct-3/4 response elements in murine metastatic bladder cancer models. Clin Cancer Res. 2008; 14:1228-38. PMID: 18281558.
  3. Lu CS, Hsieh JL, Lin CY, Tsai HW, Su BH, Shieh GS, Su YC, Lee CH, Chang MY, Wu CL, Shiau AL. Potent antitumor activity of Oct4 and hypoxia dual-regulated oncolytic adenovirus against bladder cancer. Gene Ther. 2015; 22:305-15. PMID: 25588741.

Reviewer 2 Report

This exciting manuscript examines the preclinical development of immune response modulating antitumor virus. The authors performed both in vitro experiments and used a xenograft tumor model. The manuscript can be published in its present form after minor corrections of English.

The manuscript needs minor corrections of English.

Author Response

Oct4 and hypoxia dual-regulated oncolytic adenovirus armed with shRNA targeting dendritic cell immunoreceptor exerts potent antitumor activity against bladder cancer

Hu et al. (biomedicines-2540399)

Comments on the Quality of English Language

The manuscript needs minor corrections of English.

Response:

We thank the reviewer’s comments and revise our manuscript according to the comment.

Round 2

Reviewer 1 Report

See comments in green related to  revisions

Hu et al. (biomedicines-2540399)

Point-by-point responses to the editor and reviewers:

Response to the editor: We have reworded the parts indicated by repetition check.

Comments 1. Some phrasing is not clear or confusing, especially in discussion.

Response: We apologize for lack of clarity and causing confusion We have carefully revised the manuscript on the basis of science and English language.

2.     Methods:

3.             the amount of cells used for lysates in CTL assays must be determined

Response:Thank you for your helpful suggestion. We have added the required information for the CTL assay. We included the following in Materials and Methods section in the revised version of the manuscript (page 5, lines 210-214).

“Nonadherent cells were harvested and used as effector cells. Various numbers of the splenocytes (5 × 105, 2.5 × 105, 1.25 × 105, and 6.25 × 104) were plated in 96-well micro-plates and cocultured with the target MBT-2/Luc cells (5 × 103/well) in triplicate to achieve effector-to-target (E:T) ratios of 100:1, 50:1, 25:1, and 12.5:1.”

(related to fig3). Control spleen cells must be included.

Response: Thank you for your helpful comment. Since the purpose of the CTL assay shown in Figure 3B was to compare the antitumor activity of the spleen cells collected from tumor-bearing mice that had been administered with Ad.shDCIR or Ad.LCY. The assay did not include control spleen cells obtained from normal mice. We presume that the CTL activity of control splenocytes should be minimal.

1. Reviewer: This is an unacceptable  response. They must use appropriate controls by which they can measure the specific CTL activity over the background.

1.             Number of mice per group must be provided (related to fig 4).

Response: We have added the mouse number per group in the Materials and Methods section (page 5, lines 221-222).

1.   Tumor volume formula must be checked

Response:To enhance clarity, we have revised the formula for measuring tumor volumes in the Materials and Methods section (page 5, lines 224-226). We stated “Tumors were measured every three days in two perpendicular axes with a tissue caliper. Tumor volume was calculated as (length of tumor) × (width of tumor)2 × 0.45.”

2. Reviewer: The correct formula is: length of tumor) × (width of tumor)2 × 0.52

1.   (1) Mock infections related to all figures-results must include PBS and an empty adenovirus control.

Response: Thank you for your critical comment. In Figures 1B and 1C, culture medium was used as the control. In Figure 2, mock-treated cells corresponded to uninfected cells in culture medium. In Figure 4, saline served as the vehicle control. However, in Figure 3, we did not include naïve normal splenocytes as control effector cells. The backbones of oncolytic adenoviruses, such as Ad.shDCIR and Ad.LCY, were very different from those of wild-type adenoviruses. Oncolytic adenoviruses can selectively replicate in and kill tumor cells while sparing normal cells. Wild-type adenovirus can infect a variety of normal and tumor cells, leading to cell death. Therefore, this unmodified wild-type adenovirus is not suitable to serve as control viruses. Of note, the backbone of Ad.shDCIR and Ad.LCY was identical except for the expression cassette of DCIR shRNA in Ad.shDCIR. Therefore, the two oncolytic adenoviruses are appropriate for side-by-side comparisons for the advantage of shDCIR expression in enhancing their antitumor activity.

3. Reviewer: Good practice inthis kind of studies is a. The use of at least 2 different sh molecules to increase confidence about he specificity of the results and b.   The use of a random shRNA sequence –often referred to as scrambled sh sequence to use as a base line for measuring   the target- specific affect. This scambled sh virus is a more appropriate control than the LCY parental virus.

(2) Are the viruses replicative or not in the different cell types?

Response: We have previously published several papers regarding the construction of different E1B-55 kD-deleted oncolytic adenoviruses with various promoters or response elements that are active in cancer cells but not in normal cells. These engineered adenoviruses replicate in and lyse different human bladder cancer cells,1 but not normal human urothelial cells.2,3 Importantly, we have also identified that murine MBT-2 cancer cells are permissive to adenoviral infection.4 Notably, these oncolytic adenoviruses can replicate in MBT-2 cells. Furthermore, adenoviral fiber and hexon proteins, which are two viral late protein, are detected in tumor tissue of the mice receiving oncolytic virotherapy, indicating oncolytic adenoviruses can produce viral progeny in MBT-2 cells and tumors.2,3 Therefore, MBT-2 tumor model allows us to evaluate the therapeutic efficacy of oncolytic adenoviruses in immunocompetent mice. Compared to xenograft tumor models in immunodeficient mice, syngeneic tumor models in immunocompetent mice provide the appropriate tumor microenvironment and immunologic compatibility between tumor cells and the animal that is more closely representative of the mouse tumor and also more closely exemplifies the human physiology.

References:

1.           Hsieh JL, Wu CL, Lai MD, Lee CH, Tsai CS, Shiau AL. Gene therapy for bladder cancer using E1B-55 kD-deleted adenovirus in combination with adenoviral vector encoding plasminogen kringles 1-5. Br J Cancer. 2003; 88:1492-9. PMID: 12778082.

2.          Chang CC, Shieh GS, Wu P, Lin CC, Shiau AL, Wu CL. Oct-3/4 expression reflects tumor progression and regulates motility of bladder cancer cells. Cancer Res. 2008; 68:6281-91. PMID: 18676852.

3.          Wu CL, Shieh GS, Chang CC, Yo YT, Su CH, Chang MY, Huang YH, Wu P, Shiau AL. Tumor-selective replication of an oncolytic adenovirus carrying oct-3/4 response elements in murine metastatic bladder cancer models. Clin Cancer Res. 2008; 14:1228-38. PMID: 18281558.

4.             Shieh GS, Shiau AL, Yo YT, Lin PR, Chang CC, Tzai TS, Wu CL. Low-dose etoposide enhances telomerase-dependent adenovirus-mediated cytosine deaminase gene therapy through augmentation of adenoviral infection and transgene expression in a syngeneic bladder tumor model. Cancer Res. 2006; 66:9957-66. PMID: 17047058.

3.      Results :

Fig 1: They measure RNA, but protein levels are more informative for this king of experiment.

Response: In Figure 1, we detected mRNA levels of DCIR in bone marrow-derived dendritic cells (BMDCs) after treatment with the conditioned medium of MBT-2 cells that had been infected with oncolytic adenoviruses. Since shRNAs generally interact with the 3’-UTR regions of their targets, resulting in mRNA cleavage, we examined the effect of Ad.shDCIR on downregulation of Clec4a mRNA expression with RT-qPCR. Thus, we did not detect the protein levels of the treated cells. Nevertheless, we have used A549/DCIR-c-myc cells to examine Clec4a protein levels in cells after infection with Ad.shDCIR by immunoblot analysis with anti-c-myc antibody (Table 1). We have added the following in the Results section of the revised version of the manuscript (page 5, lines 241-243).

“We first used A549/DCIR-c-myc cells to examine the silencing effect of Ad.shDCIR. The expression of DCIR was reduced in cells infected with Ad.shDCIR as examined by immunoblot analysis (data not shown).”

Table 1. Knockdown efficacy of Ad.shDCIR in reducing DCIR-c-myc protein expression. A549 cells were transduced with lentiviral vectors encoding DCIR-c-myc fusion proteins to generate A549/DCIR-c-myc cells. The cells were infected with Ad.shDCIR for 48 h, and their DCIR expression was examined by immunoblot analysis with anti-c-myc antibody.

The conclusion that DCs ate not directly infectable by adenovirus is important. Is this true for human DC also.

Response: Since Ad.shDCIR cannot replicate and induce cytolytic effects in normal cells, it fails to silence DCIR expression in DCs. As oncolytic adenoviruses cannot replicate in normal cells, we presume that they have no noticeable effects on human DCs.

The presence of extracellular vesicles carrying the shRNA of interest is interesting but they do not provide evidence.

Response:Thank you for your constructive comment. We have added the following in the Discussion section of the revised version of the manuscript (page 9, lines 374-379):

“In the present study, we did not provide evidence to show that extracellular vesicles may carry shRNAs of interest. However, there are accumulating data showing that extracellular vesicles can be released almost ubiquitously by every living cell, including tumor cells and immune cells. They can carry various molecules, such as protein, DNA, RNA, microRNA, long non-coding RNA (lncRNA), shRNA, and more. The extracellular vesicles play roles in cancer progression and immune regulation.”

4. Reviewer: I would prefer experimental support of their claim. The easiest would be to use even crude cell –free supernatants from control and infected cells to show transfer of the sh effect  to  DCs.  

Fig2. In A,B, C , which cells are NIH3T3 and which MBT2 is confusing. The replicates are not clearly shown.

Response:We have revised the legend of Figure 2 and indicated MBT-2 and NIH 3T3 cells in A and C panels of Figure 2, respectively, in the revised version of the manuscript.

Fig 4. Where the legend, statistics, number of mice of part C (survival) of this figure?

Response:The legend for Figure 4 has been revised in the revised version of the manuscript (page 8, lines 331-338).

Results of B and C concerning Ad.LCY are contradictory. Since the LCY tumors hardly grow (similar to DCIR that however has good survival) what is the cause of death?

Response:Thank you for raising this important question. We have added the following in the Discussion section of the revised version of the manuscript (page 11, lines 454-462):

“In the present study, while Ad.shDCIR and Ad.LCY exerted similar antitumor effects on retarding tumor growth, Ad.shDCIR was superior to Ad.LCY in prolonging mouse survival. These seemingly contradictory results deserve further investigation. However, it is not necessarily expected that tumor-bearing mice would definitely die of larger tumor burdens. Furthermore, the mice may also die as a result of metastatic tumors even in small sizes. Thus, it is reasonable to speculate that Ad.shDCIR may be more potent than Ad.LCY in promoting antitumor immune responses, which may be beneficial for inhibiting tumor metastasis, leading to enhancing the survival of tumor-bearing mice. However, further studies are required to elucidate the mechanisms of action of these oncolytic adenoviruses.”

5.Reviewer: This a very important point indeed.  Why the authors have not already checked for metastasis?

Discussion an general experimental set up :

1.The part of human xenografts vs isogeneic mouse grafts is confusing.

Response:

Thank you for your critical comment. We have revised the text in the Discussion section in the revised version of the manuscript as follows (page 10, lines 431-442):

“We have previously demonstrated that either Ad.9OC (an Oct4-dependent oncolytic adenovirus driven by nine copies of the ORE) or Ad.LCY (an Oct4 and hypoxia dual-regulated oncolytic adenovirus driven by the HRE/ORE promoter) exerts oncolytic activities in mice bearing human TCCSUP tumor xenografts or murine MBT-2 syngeneic tumors. Notable, Ad.LCY exhibits higher antitumor activity than Ad.9OC in NOD/SCID mice bearing human TCCSUP bladder tumor xenografts in terms of inhibiting tumor growth and prolonging mouse survival [12]. In C3H/HeN mice bearing murine MBT-2 syngeneic tumors, Ad.LCY is more effective in inhibiting tumor growth; however, the two viruses exert similar effects on extending mouse survival [12]. Since these two oncolytic adenoviruses exploit tumor properties of Oct4 overexpression and hypoxia for designing their tumor selectivity and efficacy, they exert similar antitumor effects in immunocompetent mice [12].”

2.   To determine if there is an immune component in what they observed they should perform parallel experiments in immunocompromised mice.

Response:Thank you for your valuable comment. We have included the text in the Discussion section of the revised version of the manuscript as follows (page 10, lines 442-453).

“Xenograft mouse models of human tumors grown in immunocompromised mice, such as NOD/SCID mice, are wildly used for studying cancer metastasis and anticancer drug screening. The xenograft model does not mimic the natural immune response of the human body due to immunodeficiency. By contrast, syngeneic tumor models in immunocompetent mice provide the appropriate tumor microenvironment and immunologic compatibility between tumor cells and the mice that is more closely representative of the mouse tumor and also more closely exemplifies human physiology. In the current study, we employed strategies of tumor-selective virotherapy and silencing of the negative immune regulator Clec4a to construct Ad.shDCIR. Since this novel engineered adenovirus combines oncolytic virotherapy and immunotherapy, it is more appropriate to use syngeneic MBT-2 tumor model in immunocompetent C3H/HeN mice for evaluating its antitumor efficacy.

6. Reviewer:  The point here was to compare survival and growth of the MBT-2  cells  growing in parallel in  immunocompetent  and immunocompromised mice.

3.             Their data may simply reflect an empty virus effect in cell survival

Response: Thank you for your valuable comment. Originally, we constructed Ad5WS1, which is an E1B-55 kD-deleted oncolytic adenovirus derived from human adenovirus type 5, driven by the adenoviral E1A promoter.1 Subsequently, we replaced the E1A promoter in the viral backbone with the Oct4 response element (ORE) and the hypoxia response element (HRE) in conjunction with the ORE to generate Ad.9OC and Ad.LCY targeting Oct4 (stemness) and hypoxia in tumors, respectively.2,3 In the present study, we further inserted the expression cassette of shClec4a driven the U6 promoter upstream the HRE/ORE hybrid promoter to create Ad.shDCIR. Considering the complexity of these engineered adenoviruses, it is not suitable to use an empty virus, either wild-type human adenovirus type 5 or replication-defective adenoviral vector frequently used as a gene therapy or vaccine vector, as the control virus. On the basis of our previous reports, our oncolytic adenoviruses indeed exert tumor-selective cytolytic activities in vitro and oncolytic activities in mouse tumor models.

7. Reviewer:  This is related to  point 3 above. The question here is to  use an appropriate control for the extra elements that the engineered virus has relative to its parent as It is common practice in genetic experimentation of this type ie include a virus that does not contains U6 driving  sh-scrambled  carrying virus.  

often confusing  phrasing .Need major editing for clarity 

Author Response

Oct4 and hypoxia dual-regulated oncolytic adenovirus armed with shRNA targeting dendritic cell immunoreceptor exerts potent antitumor activity against bladder cancer

Hu et al. (biomedicines-2540399)

Point-by-point second responses to Reviewer 1

  1. Methods:
  2. the amount of cells used for lysates in CTL assays must be determined

Response: Thank you for your helpful suggestion. We have added the required information for the CTL assay. We included the following in Materials and Methods section in the revised version of the manuscript (page 5, lines 210-214).

“Nonadherent cells were harvested and used as effector cells. Various numbers of the splenocytes (5 × 105, 2.5 × 105, 1.25 × 105, and 6.25 × 104) were plated in 96-well micro-plates and cocultured with the target MBT-2/Luc cells (5 × 103/well) in triplicate to achieve effector-to-target (E:T) ratios of 100:1, 50:1, 25:1, and 12.5:1.”

(related to fig 3). Control spleen cells must be included.

Response: Thank you for your helpful comment. Since the purpose of the CTL assay shown in Figure 3B was to compare the antitumor activity of the spleen cells collected from tumor-bearing mice that had been administered with Ad.shDCIR or Ad.LCY. The assay did not include control spleen cells obtained from normal mice. We presume that the CTL activity of control splenocytes should be minimal.

  1. Reviewer: This is an unacceptable  response. They must use appropriate controls by which they can measure the specific CTL activity over the background.

Second response:

We agree with the reviewer’s comment. In our previous data published in Molecular Therapy Nucleic Acids1 (ref. 7), we performed the same CTL assay using the same target cells (MBT-2 cells expressing luciferase) (Figure 3). In this paper, we included effectors cells isolated from both the control mice and vector control mice. Our results revealed that splenocytes from mice receiving shRNA targeting Clec4a (Clec4a-1 or Clec4a-3) induced stronger CTL responses than those from mice receiving a vector plasmid (pLKO_AS1) or no treatment (control). These two control splenocytes had similar basal levels of CTL activities. Since the present study was the extension of our previous work1, we expect that the CTL activity of the two control splenocytes should be minimal. We have included this part in the Results section (page 7, lines 305-312).

In normal situations, it is easy for us to perform the experiment requested by the reviewers. However, since we have closed the laboratory due to retirement, experiments, in particularly animal experiments, become very difficult to carry out. It is also worth mentioning that regulation of the Laboratory Animal Care and Use Committee of our university follows the 3Rs (replacement, reduction, and refinement) for animal welfare. We expect that an application for a new animal experiment for the purpose of including control mice may take times and be disapproved. We apologize for not able to perform the experiment suggested by the reviewer.

Reference:

  1. Weng TY, Li CJ, Li CY, Hung YH, Yen MC, Chang YW, Chen YH, Chen YL, Hsu HP, Chang JY, Lai MD. Skin delivery of Clec4a small hairpin RNA elicited an effective antitumor response by enhancing CD8+ immunity in vivo. Mol Ther Nucleic Acids. 2017; 9:419-27. PMID: 29246320.

  1. Number of mice per group must be provided (related to fig 4).

Response: We have added the mouse number per group in the Materials and Methods section (page 5, lines 221-222).

  1. Tumor volume formula must be checked

Response: To enhance clarity, we have revised the formula for measuring tumor volumes in the Materials and Methods section (page 5, lines 224-226). We stated “Tumors were measured every three days in two perpendicular axes with a tissue caliper. Tumor volume was calculated as (length of tumor) × (width of tumor)2 × 0.45.”

  1. Reviewer: The correct formula is: length of tumor) × (width of tumor)2× 0.52

Second response:

Thank you for your correction. We have revised the formula for measuring tumor volumes and cited a new reference (page 5, lines 225-226, ref. 17). The tumor volumes have also been corrected.

  1. (1) Mock infections related to all figures-results must include PBS and an empty adenovirus control.

Response: Thank you for your critical comment. In Figures 1B and 1C, culture medium was used as the control. In Figure 2, mock-treated cells corresponded to uninfected cells in culture medium. In Figure 4, saline served as the vehicle control. However, in Figure 3, we did not include naïve normal splenocytes as control effector cells. The backbones of oncolytic adenoviruses, such as Ad.shDCIR and Ad.LCY, were very different from those of wild-type adenoviruses. Oncolytic adenoviruses can selectively replicate in and kill tumor cells while sparing normal cells. Wild-type adenovirus can infect a variety of normal and tumor cells, leading to cell death. Therefore, this unmodified wild-type adenovirus is not suitable to serve as control viruses. Of note, the backbone of Ad.shDCIR and Ad.LCY was identical except for the expression cassette of DCIR shRNA in Ad.shDCIR. Therefore, the two oncolytic adenoviruses are appropriate for side-by-side comparisons for the advantage of shDCIR expression in enhancing their antitumor activity.

  1. Reviewer:Good practice in this kind of studies is a. The use of at least 2 different sh molecules to increase confidence about the specificity of the results and b.   The use of a random shRNA sequence –often referred to as scrambled sh sequence to use as a base line for measuring   the target- specific affect. This scrambled sh virus is a more appropriate control than the LCY parental virus.

Second response:

Thank you for your critical comments. Since Ad.shDCIR was derived from Ad.LCY, we compared the antitumor activities of these two oncolytic viruses, rather than using Ad.LCY as the control virus. The present study was the extension of our previous work (ref. 7). Previously, we screened four different shRNA sequences specific to mouse Clec4a and found that three of the four sequences, designated shClec4a-1 (targeting the Clec4a2 coding sequence), shClec4a-3 (targeting the 3’ UTR of Clec4a2 mRNA), and shClec4a (targeting the Clec4a2 coding sequence) could efficiently knock down the expression of Clec4a and significantly delayed MBT-2 tumor growth in mice via gene gun-mediated skin delivery of plasmid DNA carrying shRNA targeting Clec4a. Both shClec4a-1 and shClec4a2, which targeted different sequences of the Clec4a2 coding region, had similar knockdown efficiency on Clec4a2 expression and antitumor activities in the MBT-2 tumor model. We also excluded the off-target effects of Clec4a2 shRNA by the compensation experiment. The expression of Clec4a was compensated by co-transfection of 293T cells with shClec4a-3 (targeting the 3’ UTR) and Clec4a2-myc expression plasmid (without the 3’ UTR). Notably, we used pLKO_AS1, which contains the U6 promoter with no shRNA sequence, as the empty vector control. This is a backbone lentiviral transfer vector for inserting shRNA sequences. Although we did not use the scrambled shRNA sequence as the control shRNA for our previous experiments, the shClec4a-2 that targeted the Clec4a2 coding sequence failed to knock down the Clec4a2-myc expression and thus may also serve as another control shRNA. Accordingly, shClec4a-2 did not exhibit any antitumor effect. On the basis of our previous reports, the aim of the present work was to combine the two strategies that we have already reported1-3, namely the Oct4 and hypoxia dual-regulated oncolytic adenovirus Ad.LCY (ref. 11 and 12) as well as Clec4a shRNA (ref. 7) for enhancing the antitumor efficacy of oncolytic adenoviruses. Since the present study was a proof of principle study, for saving time-consuming procedures for producing recombinant adenoviruses, we did not use the control viruses as suggested by the reviewers. 

References:

  1. Wu CL, Shieh GS, Chang CC, Yo YT, Su CH, Chang MY, Huang YH, Wu P, Shiau AL. Tumor-selective replication of an oncolytic adenovirus carrying oct-3/4 response elements in murine metastatic bladder cancer models. Clin Cancer Res. 2008; 14:1228-38. PMID: 18281558.
  2. Lu CS, Hsieh JL, Lin CY, Tsai HW, Su BH, Shieh GS, Su YC, Lee CH, Chang MY, Wu CL, Shiau AL. Potent antitumor activity of Oct4 and hypoxia dual-regulated oncolytic adenovirus against bladder cancer. Gene Ther. 2015; 22:305-15. PMID: 25588741.
  3. Weng TY, Li CJ, Li CY, Hung YH, Yen MC, Chang YW, Chen YH, Chen YL, Hsu HP, Chang JY, Lai MD. Skin delivery of Clec4a small hairpin RNA elicited an effective antitumor response by enhancing CD8+ immunity in vivo. Mol Ther Nucleic Acids. 2017; 9:419-27. PMID: 29246320.

=====================================================================

The presence of extracellular vesicles carrying the shRNA of interest is interesting but they do not provide evidence.

Response: Thank you for your constructive comment. We have added the following in the Discussion section of the revised version of the manuscript (page 9, lines 374-379):

“In the present study, we did not provide evidence to show that extracellular vesicles may carry shRNAs of interest. However, there are accumulating data showing that extracellular vesicles can be released almost ubiquitously by every living cell, including tumor cells and immune cells. They can carry various molecules, such as protein, DNA, RNA, microRNA, long non-coding RNA (lncRNA), shRNA, and more. The extracellular vesicles play roles in cancer progression and immune regulation.”

  1. Reviewer: I would prefer experimental support of their claim. The easiest would be to use even crude cell –free supernatants from control and infected cells to show transfer of the sh effect to DCs.

Second response:

As we have no experimental support for this claim described in the Discussion section, we have deleted the above text in the revised version of the manuscript.

=====================================================================

Fig 4. Where the legend, statistics, number of mice of part C (survival) of this figure?

Response: The legend for Figure 4 has been revised in the revised version of the manuscript (page 8, lines 331-338).

Results of B and C concerning Ad.LCY are contradictory. Since the LCY tumors hardly grow (similar to DCIR that however has good survival) what is the cause of death?

Response: Thank you for raising this important question. We have added the following in the Discussion section of the revised version of the manuscript (page 11, lines 454-462):

“In the present study, while Ad.shDCIR and Ad.LCY exerted similar antitumor effects on retarding tumor growth, Ad.shDCIR was superior to Ad.LCY in prolonging mouse survival. These seemingly contradictory results deserve further investigation. However, it is not necessarily expected that tumor-bearing mice would definitely die of larger tumor burdens. Furthermore, the mice may also die as a result of metastatic tumors even in small sizes. Thus, it is reasonable to speculate that Ad.shDCIR may be more potent than Ad.LCY in promoting antitumor immune responses, which may be beneficial for inhibiting tumor metastasis, leading to enhancing the survival of tumor-bearing mice. However, further studies are required to elucidate the mechanisms of action of these oncolytic adenoviruses.”

  1. Reviewer: This a very important point indeed.  Why the authors have not already checked for metastasis?

Second response:

Thank you for your critical comment. Indeed, this is an important issue. However, in this study, we used a subcutaneous MBT-2 tumor model for easy identification of the antitumor effects of oncolytic adenoviruses. An experimental lung metastatic tumor model is frequently employed by intravenous injection of tumor cells into mice via the tail vein. Thus, metastatic lung tumor nodules can be observed and quantified for metastasis. However, these experiments are beyond the scope of the current work. Therefore, we discussed the possibility of metastasis. As we mentioned, further animal studies are required to elucidate the underlying mechanisms in the future.

=====================================================================

  1. To determine if there is an immune component in what they observed they should perform parallel experiments in immunocompromised mice.

Response: Thank you for your valuable comment. We have included the text in the Discussion section of the revised version of the manuscript as follows (page 10, lines 442-453).

“Xenograft mouse models of human tumors grown in immunocompromised mice, such as NOD/SCID mice, are widely used for studying cancer metastasis and anticancer drug screening. The xenograft model does not mimic the natural immune response of the human body due to immunodeficiency. By contrast, syngeneic tumor models in immunocompetent mice provide the appropriate tumor microenvironment and immunologic compatibility between tumor cells and the mice that is more closely representative of the mouse tumor and also more closely exemplifies human physiology. In the current study, we employed strategies of tumor-selective virotherapy and silencing of the negative immune regulator Clec4a to construct Ad.shDCIR. Since this novel engineered adenovirus combines oncolytic virotherapy and immunotherapy, it is more appropriate to use syngeneic MBT-2 tumor model in immunocompetent C3H/HeN mice for evaluating its antitumor efficacy.

  1. Reviewer: The point here was to compare survival and growth of the MBT-2 cells growing in parallel in immunocompetent and immunocompromised mice.

Second response:

Thank you for your comment. Oncolytic viruses can specifically kill cancer cells and induce antitumor immune responses. We have discovered that the MBT-2 tumor growing in syngeneic immunocompetent C3H/HeN mice is a very appropriate bladder tumor model for evaluating the antitumor activities of oncolytic adenoviruses1,2. As suggested by the reviewer, immunocompromised mice bearing human xenograft tumors can be tested in parallel. However, based on our previous studies, we found that syngeneic tumor models resemble more closely to the real-life conditions. More importantly, the syngeneic MBT-2 tumor model in immunocompetent mice is more suitable for evaluating the antitumor activities of our novel Ad.shDCIR oncolytic adenovirus targeting the mouse DC immunoreceptor (DCIR) Clec4a2. Since the interactions of cancer cells with the immune system are complex, preclinical studies of DC-related immunotherapies require specific animal models that fully resemble human cancer in clinical settings. The xenograft model does not mimic the natural immune response of the human body due to the immunodeficiency of the mice. By contrast, syngeneic tumor models in immunocompetent mice provide the appropriate tumor microenvironment and immunologic compatibility between tumor cells and the mice that is more closely representative of the mouse tumor and also more closely exemplifies human physiology. Besides, an additional experiment that uses immunocompromised mice in parallel with immunocompetent mice for comparison of antitumor activities of oncolytic adenoviruses may not be approved by the Laboratory Animal Care and Use Committee of our university due to the 3Rs principles in animal welfare.   

References

  1. Shieh GS, Shiau AL, Yo YT, Lin PR, Chang CC, Tzai TS, Wu CL. Low-dose etoposide enhances telomerase-dependent adenovirus-mediated cytosine deaminase gene therapy through augmentation of adenoviral infection and transgene expression in a syngeneic bladder tumor model. Cancer Res. 2006; 66:9957-66. PMID: 17047058.
  2. Wu CL, Shieh GS, Chang CC, Yo YT, Su CH, Chang MY, Huang YH, Wu P, Shiau AL. Tumor-selective replication of an oncolytic adenovirus carrying oct-3/4 response elements in murine metastatic bladder cancer models. Clin Cancer Res. 2008; 14:1228-38. PMID: 18281558.

  1. Their data may simply reflect an empty virus effect in cell survival

Response: Thank you for your valuable comment. Originally, we constructed Ad5WS1, which is an E1B-55 kD-deleted oncolytic adenovirus derived from human adenovirus type 5, driven by the adenoviral E1A promoter.1 Subsequently, we replaced the E1A promoter in the viral backbone with the Oct4 response element (ORE) and the hypoxia response element (HRE) in conjunction with the ORE to generate Ad.9OC and Ad.LCY targeting Oct4 (stemness) and hypoxia in tumors, respectively.2,3 In the present study, we further inserted the expression cassette of shClec4a driven the U6 promoter upstream the HRE/ORE hybrid promoter to create Ad.shDCIR. Considering the complexity of these engineered adenoviruses, it is not suitable to use an empty virus, either wild-type human adenovirus type 5 or replication-defective adenoviral vector frequently used as a gene therapy or vaccine vector, as the control virus. On the basis of our previous reports, our oncolytic adenoviruses indeed exert tumor-selective cytolytic activities in vitro and oncolytic activities in mouse tumor models.

References:

  1. Hsieh JL, Wu CL, Lee CH, Shiau AL. Hepatitis B virus X protein sensitizes hepatocellular carcinoma cells to cytolysis induced by E1B-deleted adenovirus through the disruption of p53 function. Clin Cancer Res. 2003; 9:338-45. PMID: 12538486.
  2. Wu CL, Shieh GS, Chang CC, Yo YT, Su CH, Chang MY, Huang YH, Wu P, Shiau AL. Tumor-selective replication of an oncolytic adenovirus carrying oct-3/4 response elements in murine metastatic bladder cancer models. Clin Cancer Res. 2008; 14:1228-38. PMID: 18281558.
  3. Lu CS, Hsieh JL, Lin CY, Tsai HW, Su BH, Shieh GS, Su YC, Lee CH, Chang MY, Wu CL, Shiau AL. Potent antitumor activity of Oct4 and hypoxia dual-regulated oncolytic adenovirus against bladder cancer. Gene Ther. 2015; 22:305-15. PMID: 25588741.

  1. Reviewer:  This is related to  point 3 above. The question here is to use an appropriate control for the extra elements that the engineered virus has relative to its parent as it is common practice in genetic experimentation of this type ie include a virus that does not contains U6 driving sh-scrambled  carrying virus.

Second response:

Thank you for your valuable comments. Although we did not generate an adenovirus that did not contain U6 driven sh-scrambled sequence, we actually used pLKO_AS1 plasmid as the empty control vector in our previous work (ref. 7). This plasmid is an shRNA lentiviral transfer vector that contains the U6 promoter and two BsmB1 recognition sites for easy insertion of BsmB1-restricted shRNA sequences generated by PCR into the vector. Our data have shown that the empty control vector could not silence Clec4a expression, nor could it exert antitumor activities. Thus, it served as a suitable control vector. Ideally, generation of different control viruses for parallel comparisons is highly warranted. However, due to the limitation of our resources, we were unable to accomplish such experiments.

Round 3

Reviewer 1 Report

Following revisions I have no further comments